# Antimicrobial Peptide Conjugated on Graphene Oxide-Containing Sulfonated Polyetheretherketone Substrate for Effective Antibacterial Activities against *Staphylococcus aureus*

**DOI:** 10.3390/antibiotics12091407

**Published:** 2023-09-05

**Authors:** Selvaraj Rajesh Kumar, Chih-Chien Hu, Truong Thi Tuong Vi, Dave W. Chen, Shingjiang Jessie Lue

**Affiliations:** 1Department of Chemical and Materials Engineering, Chang Gung University, Taoyuan City 333, Taiwan; rajeshkumarcgu@mail.cgu.edu.tw (S.R.K.); vi123456@cgmh.org.tw (T.T.T.V.); 2Department of Orthopedics, Chang Gung Memorial Hospital, Linkou, Taoyuan City 333, Taiwan; r52906154@cgmh.org.tw; 3Division of Pediatric Gastroenterology and Hepatology, Department of Pediatrics, Chang Gung Memorial Hospital, Taoyuan City 333, Taiwan; 4Department of Orthopedic Surgery, Chang Gung Memorial Hospital, Keelung City 204, Taiwan; 5Department of Safety, Health and Environment Engineering, Ming Chi University of Technology, New Taipei City 243, Taiwan

**Keywords:** antimicrobial peptide, antibacterial efficiency, biofilm resistance, nisin islets, implant surface modification

## Abstract

In the present study, the antimicrobial peptide nisin was successfully conjugated onto the surface of sulfonated polyetheretherketone (SPEEK), which was decorated with graphene oxide (GO) to investigate its biofilm resistance and antibacterial properties. The PEEK was activated with sulfuric acid, resulting in a porous structure. The GO deposition fully covered the porous SPEEK specimen. The nisin conjugation was accomplished using the crosslinker 1–ethyl–3–(3–dimethylaminopropyl)carbodiimide (EDC) through a dip-coating method. The surface micrographs of the SPEEK-GO-nisin sample indicated that nisin formed discrete islets on the flat GO surface, allowing both the GO and nisin to perform a bactericidal effect. The developed materials were tested for bactericidal efficacy against *Staphylococcus aureus* (*S. aureus*). The SPEEK-GO-nisin sample had the highest antibacterial activity with an inhibition zone diameter of 27 mm, which was larger than those of the SPEEK-nisin (19 mm) and SPEEK-GO (10 mm) samples. Conversely, no inhibitory zone was observed for the PEEK and SPEEK samples. The surface micrographs of the bacteria-loaded SPEEK-GO-nisin sample demonstrated no bacterial adhesion and no biofilm formation. The SPEEK-nisin and SPEEK-GO samples showed some bacterial attachment, whereas the pure PEEK and SPEEK samples had abundant bacterial colonies and thick biofilm formation. These results confirmed the good biofilm resistance and antibacterial efficacy of the SPEEK-GO-nisin sample, which is promising for implantable orthopedic applications.

## 1. Introduction

Microbial contamination poses a significant threat to the food and healthcare industries, with potential direct impacts on people’s health [1,2]. Thus, it is imperative to prioritize the development of novel antibacterial agents that can effectively prevent bacterial growth [3]. Although antibiotics have shown success in many cases, the overuse of these drugs and the inability of conventional therapies to eradicate bacterial infections have given rise to an escalating threat of antibiotic resistance among bacterial populations [4]. Therefore, finding novel antibiotic prospects is also crucial due to the rising bacterial resistance to current antibiotics. To tackle these issues, the development of novel antimicrobial peptides (AMPs) resulted in finding a family of anti-infective agents, which have been considered in recent decades as a promising substitute for conventional antibiotics. These AMPs exhibit an antibacterial mechanism through their interaction with bacterial cell membranes, which result in changes to membrane permeability [5,6]. As a result, AMP-related agents are particularly favorable for promising antibacterial agents and cell compatibility [7]. In addition, these AMPs could strongly prevent biofilm formation through resisting bacteria on the surfaces of the prepared artificial composites employed in orthopedics [8,9].

Polyetheretherketone (PEEK) is a semi-crystalline thermoplastic polymer that exhibits excellent thermal, chemical, mechanical, and environmental resistance [10]. PEEK is not easily degraded in physiological/biological environments, and it does not release any toxic ingredients owing to its good corrosion resistance and stability. Specifically, the elastic modulus of PEEK is similar to that of human cortical bone [11]. Additionally, PEEK implants can effectively support body weight without causing side effects such as stiffness, reduced elasticity, compromised tensile strength, or susceptibility to abrasion, distortion, and fatigue, thus enabling them to coexist harmoniously with human bone [12,13]. Li and Zenobi et al. utilized the solid phase of PEEK from the hydroxylated metabolites of polycyclic aromatic hydrocarbons that were obtained from various matrices (acid/enzymatic hydrolysis solution, urine, etc.) [14]. Zheng et al. tailored the PEEK surface with various functional groups including –OH, –COOH, and –PO_4_H_2_ for the improvement of in vitro and in vivo osteogenic activities [15]. Although PEEK has a lower free surface energy and an inert hydrophobic interface, which results in poor binding qualities between PEEK and biomolecules, the pure PEEK substrate is associated with significant biological issues, such as limited tissue adhesion, reduced cell proliferation, bio-inertness, and elevated biofilm formation [16,17]. The absence of reactive functional groups on PEEK limits its ability to form chemical bonds with surrounding tissue, leading to poor osseointegration and device stability. This can result in implant loosening or fracture, leading to implant failure [18]. Furthermore, a significant proportion of PEEK implant failures can be attributed to bacterial growth, which lead to the development of microbial infections [17]. To mitigate these issues, the surface of PEEK needs to be modified with various biocompatible materials, and suitable antimicrobial peptide (AMP) conjugation may possibly enhance the antimicrobial activity and osseointegration for high-quality implantable biomaterials.

Nisin is a cationic polypeptide that contains the amphiphilic groups, with hydrophilic residues at the C–terminal and hydrophobic residues at the N–terminal, and is approved by the US Food and Drug Administration (FDA, 1988) [19,20]. Nisin has two different mechanisms for its antibacterial activity, including pore formation and lipid trapping. The lipid trapping mechanism occurs when the N–terminal residues (1–12) attach to the pyrophosphate moiety of lipid II, which contains the peptidoglycan molecule and interferes with bacterial cell wall production. In another mechanism, the C–terminus of nisin also promotes the pore-forming process in the surface of the bacterial cell membrane, which leads to cell leakage and eventual cell death [21,22]. The antibacterial efficacy of pure nisin against *S. aureus* is documented in the literature [23,24,25]. Aveyard et al. [5] compared the antibacterial efficiency against *Staphylococcus aureus* (*S. aureus*) and *Listeria monocytogenes* bacteria by using the covalent and electrostatic immobilization of nisin peptide on a polystyrene substrate. Following the results, it was found that the covalently immobilized nisin was significantly more effective at killing the *S. aureus* bacteria than the polymer with electrostatic immobilization. In our previous report, we found that physically or chemically (via EDC coupling) grafted AMPs on the dense PEEK surface enhance high antibacterial activity with good biofilm resistance [26]. Therefore, AMP coatings on metal or polymeric substrates show promising antibacterial activities and biofilm resistances.

The main objective of the current research is to focus on a facile method of nisin conjugation on the SPEEK substrate, as well as to investigate its antibacterial activities, along with its biofilm resistance. The etching of the PEEK surfaces, achieved via the sulfonation process, lead to the uniform formation of a three-dimensional SPEEK porous structure, thereby resulting in increased hydrophobicity. The GO grafting on the SPEEK improves the functional groups of a basal surface and hydrophilicity. Then, the AMP of nisin was conjugated on SPEEK-GO through the ethyl-3-(3-dimethylaminopropyl)carbodiimide (EDC) cross-linking agent. The morphological, physicochemical, and surface properties were studied in detail to evaluate the GO grafting and nisin conjugation on the SPEEK substrate. The SPEEK-GO-nisin-treated *S. aureus* bacteria were used to evaluate its antibacterial activities and biofilm resistance when compared with SPEEK-GO and SPEEK substrates. To the best of our knowledge, the present work is the preliminary work of successful nisin coating on SPEEK-GO surfaces, as well as an investigation of its antibacterial effects. This study presents a suitable option for a peptide grafting method of PEEK, which aids in understanding its antibacterial activities with high biofilm resistance.

## 2. Results and Discussion

The PEEK pretreatment with sulfuric acid leads to the uniform, three-dimensional porous framework in the SPEEK substrate. The porous structure was generated due to the etching process when in contact with a strong acidic medium. As a result of acid treatment, new carbon–oxygen molecules were produced, which increased the number of functional groups that enhance the chemical bonding and physical adhesive systems [27,28]. Through this, the PEEK’s bonding capability was expanded and its π–π bonds were revealed after sulfonation. In the SPEEK-GO substrate, the strong π–π stacking bonds between the GO sheet and SPEEK manifest uniform GO grafting fully covered the porous structure [29,30]. The strong adhesion between the GO layer and the SPEEK structure may have been caused by the alteration of the PEEK’s non-covalent interactions after its sulfonation process [31]. The carboxylic- and hydroxyl-rich functional groups of GO in the SPEEK-GO substrate enhanced nisin conjugation through the EDC crosslinker. In detail, the coupling chemistry involved the reaction between the –CO_2_H group of GO and the –NH_2_ group of nisin, which formed an amide bond. This covalent attachment was facilitated by the use of a cross-linking agent called EDC, which activated the carboxyl group of GO to react with the amino group of nisin [32].

### 2.1. Surface Morphological Analysis

The surface micrographs of the SPEEK, SPEEK-GO, and SPEEK-GO-nisin substrates were studied with a field emission scanning electron microscope (FESEM), as displayed in Figure 1a–f. Pure PEEK has a smooth surface without any defects, as displayed in Figure 1a,b. The SPEEK surface has a complex fiber network with a three-dimensional, sponge-like porous structure (Figure 1c,d). This porosity was obtained from the etching of the sulfonation process in the PEEK substrate. The three-dimensional porous, rough SPEEK supported the anchoring sites for further surface functionalization, which is advantageous to carbonaceous or biomolecule attachments [33,34].

The SPEEK-GO sample had a smooth surface with a wrinkled-like sheet coverage, as represented in Figure 1e,f. The grafted GO was also located deep within the three-dimensional porous structure, proving that the GO sheet was distributed on the SPEEK substrate. This indicated that the GO grafting had high impacts on the surface properties of PEEK. In the case of the SPEEK-nisin (Figure 1g,h) and SPEEK-GO-nisin (Figure 1i,j) samples, EDC was used to conjugate nisin [26] to improve AMP graft efficiency. The SPEEK-nisin sample displayed aggregated cubic-shaped nisin particles distributed on the porous surface. The SPEEK-GO-nisin sample also had non-uniform cubic-shaped nisin particles that were distributed on the flat SPEEK-GO surface, as displayed in Figure 1i,j.

The EDX analysis was utilized to study the elemental composition and the corresponding weight percentages of PEEK, SPEEK, SPEEK-GO, and SPEEK-GO-nisin samples. The EDX weight percentage of pure PEEK showed 86.63% of carbon (C) and 11.37% of oxygen (O) contents. This elemental content of pure PEEK was closely matched with a previous literature report [26]. The EDX mapping of SPEEK and SPEEK-GO samples are represented in the Appendix A. The weight percentages of C, O, and sulfur (S) contents in the SPEEK substrate were 84.55%, 13.43%, and 2.02%, respectively. The presence of S and the slight increase in O percentage, when compared with the pure PEEK, can be attributed to the sulfuric acid treatment. The weight percentage of the SPEEK-GO substrate showed lower C (79.84%) and S (0.83%) contents, whereas the oxygen weight percentage increased to 19.33%. The increased oxygen level and decreased sulfur content were due to the GO coverage on the porous SPEEK substrate. In the case of SPEEK-GO-nisin, the weight percentages of C, O, nitrogen (N), and sulfur (S) were 61.96%, 27. 73%, 9.84%, and 0.47%, respectively. The corresponding distribution ratio of those elementals were clearly displayed in the EDX mapping micrographs, as represented in Figure 2a–f. The presence of the N signal in the SPEEK-GO-nisin substrate served as evidence of successful nisin coatings, whereas the decreased S content indicated slight X-ray ejections from the background (SPEEK-GO) substrate. These EDX data further confirmed the successful coating of nisin on the surface of the SPEEK-GO substrate. Based on the nitrogen content, we estimated the nisin grafting level was tens of micrograms per square centimeter when compared to a previous report [26].

### 2.2. Structural and Contact Angle Studies

The X-ray diffraction analysis of SPEEK, SPEEK-GO, and SPEEK-GO-nisin substrates are represented in Figure 3. The four distinct plans of (110), (113), (200), and (213) correspond to the diffraction angle at 2θ of 18.7°, 20.7°, 22.8°, and 28.8°, respectively, thereby confirming the semi-crystalline behavior of PEEK, as well as matching with the standard data [30,35]. All the samples exhibited these four diffraction peaks, even after loading with GO or nisin on the SPEEK surface. However, the XRD peak intensities of SPEEK-GO and SPEEK-GO-nisin were slightly decreased due to the GO grafting and nisin conjugation effects. The results confirm that the presence of GO or nisin on the SPEEK surface does not impact the crystalline structure of the PEEK as there was no peak shift in the diffraction angles. The GO diffraction peak was essentially undetectable in both SPEEK-GO and SPEEK-GO-nisin substrates because of the low GO concentration grafted on the SPEEK surface.

The functional groups of PEEK, SPEEK, SPEEK-GO and SPEEK-GO-nisin substrates were determined by FTIR analysis, as represented in Figure 4. The PEEK spectra exhibited distinctive peaks at 673, 762, 832, and 954 cm^−1^, which corresponded to an aromatic ring and are indicative of the C–H out-of-plane bending vibrations. The broad and sharp peaks at 925 and 1304 cm^−1^ represent the symmetric stretching and bending vibration of the Aryl–(C=O)–Aryl group. The band at 1224 and 1190 cm^−1^ was assigned to the C–O–C group of asymmetric stretching vibrations. The sharp intensity peaks at 1591, 1487, 1413, and 1156 cm^−1^ belong to the vibrations of the benzene skeleton, as well as the asymmetric stretching and rocking modes. The peak at 1649 cm^−1^ was attributed to the stretching vibration of the carbonyl group. All the corresponding FTIR peaks of pure PEEK were exactly matched with previous literature reports [36,37,38]. For the SPEEK sample, the peak at around 1028 cm^−1^ represented the sulfonic acid groups that were obtained from the etching routine of PEEK. In the case of SPEEK-GO, the intense peak at 1730 cm^−1^ was attributed to the presence of -C=O stretching vibrations, which correspond to the carbonyl group from GO. Also, the small intense peak at 3618 cm^−1^ represents the O–H stretching vibration of the hydroxyl groups, which shifted toward the higher wavenumber. This shift was due to the intermolecular hydrogen bonding between SPEEK and GO. These FTIR bands further confirmed the GO grafting on the SPEEK substrate.

The covalent conjugation of the nisin peptide with SPEEK-GO was confirmed by the presence of FTIR peaks at 800–3400 cm^−1^. The small FTIR peak at 1064 cm^−1^ represents the stretching vibration of the C–O–C group. The intense peaks observed at 1645 cm^−1^ and 1560 cm^−1^ were indicative of the C=O stretching vibrations and in-plane N–H bending vibrations of the nisin peptide, which corresponded to the amide I and amide II bonds [6,39]. However, the amide band I of nisin (1650 cm^−1^) [40] underwent a slight shift to a lower wavenumber, which confirmed the intermolecular interaction between the GO and nisin peptide. The FTIR peak intensities of the corresponding PEEK bands were decreased in the SPEEK-GO-nisin sample due to the GO-nisin grafting, which occurred due to a lowering of the characteristic peak intensities from the PEEK. The presence of three absorption bands at 2875, 2928, and 2970 cm^−1^ can be attributed to the symmetric and antisymmetric C–H stretching vibrations of the peptide [41]. The broad and strong FTIR peak at 3358 cm^–1^, which was attributed to N–H stretching vibrations due to the presence of peptide amide A band from nisin, confirmed that the chemical conjugated with SPEEK-GO. Similar findings were observed by Kanchanapally et al., who grafted nisin onto a GO-based membrane [32]. Here, we confirmed the successful covalent conjugation of a nisin peptide on the PEEK surface.

The surface hydrophilic properties of the PEEK, SPEEK, SPEEK-GO, and SPEEK-GO-nisin samples were studied by measuring the contact angle via the sessile drop method, as shown in Figure 5. The water contact angle of the pure PEEK was 91°. When PEEK was treated with sulfuric acid, the contact angle was increased to 99°, this was probably due to the increase in surface roughness [34]. Based on the Cassie wetting theory, it is proposed that the surface becomes less water wettable due to the presence of microscopic patterns that trap air [38]. Such behavior was observed in the porous SPEEK sample. Upon grafting GO onto the surface of SPEEK, the water contact angle decreased to 51°. The GO layer modulated the surface polarity due to the presence of carboxyl and hydroxyl groups, thereby reducing the contact angle and increased the hydrophilicity. The SPEEK-GO-nisin showed the lowest contact angle (24°), demonstrating the most hydrophilic behavior among the tested samples. The EDC crosslinker has the ability to activate an amidation reaction between the carboxyl groups of GO and nisin peptide to facilitate the successful covalent conjugation effect [32,42]. The increased hydrophilicity observed in the SPEEK-GO-nisin surface may be attributed to the presence of the highly hydrophilic peptide coating, along with the hygroscopic GO layer. Similar behavior regarding increased hydrophilicity via the covalent immobilization of nisin on the polymeric substrate was reported elsewhere [5].

### 2.3. Antibacterial Activities against S. aureus Bacteria

The zone inhibition method was used to determine the antibacterial activity against *S. aureus* bacteria, and the presence of an inhibition zone (i.e., diameter in mm) served as an indicator of the antibacterial behavior of the samples. The pure PEEK and SPEEK samples treated with *S. aureus* bacteria displayed negligible inhibition, as shown in Figure 6. The high hydrophobicity of the PEEK and SPEEK samples can limit their ability to interact with bacterial cell walls, thus leading to minimal inhibition and increased bacterial colonization on their surfaces. In contrast, the SPEEK-GO sample exhibited a notable inhibition zone of 10 mm. The reactive oxygen species from the GO induced oxidation stress to the bacteria, as well as the expression of some antibacterial activity [43,44].

The SPEEK-nisin sample showed a larger inhibition zone of 19 mm, which was higher than that observed in the SPEEK-GO sample. Nisin is known as an effective AMP and shows good biocompatibility. It is proposed that the positively charged nisin can adhere to the bacteria cell wall (which has a negative charge), as well as cause pores or shrinkage [26]. The presence of nisin leads to a distortion of the inner membrane through non-specific interactions and further restrictions of the bacterial growth phases [45]. Many researchers found that nisin has strong antibacterial activity against Gram-positive bacteria because of its capacity to attach to the lipid-II components of cell membranes [40,46,47]. The SPEEK-GO-nisin sample, among the samples against *S. aureus,* exhibited the largest inhibition zone of 27 mm. Such a finding demonstrates the synergistic bactericidal efficacy achieved by combining a continuous GO layer and discrete nisin islets (Figure 1i,j). Recently, Jiang et al. reported that nisin-grafted magnetic GO nanohybrids exhibit good antibacterial activities against *S. aureus* due to the severe membrane disruptions from GO. In addition, nisin contributes to the identical protein leakage mechanism [6]. Notably, the nisin and nisin-GO coatings not only improved antibacterial performance, but also maintained good cell viability [6,47,48,49].

### 2.4. *Bacteria Adhered on Samples*

The surface micrographs of the post bacterial analysis of *S. aureus* when using the SPEEK, SPEEK-GO, SPEEK-nisin, and SPEEK-GO-nisin samples are represented in Figure 7a–f. The *S. aureus* bacteria fully occupied the dense and porous spaces of the pure PEEK and SPEEK samples, resulting in a high-density level of biofilm, as shown in Figure 7a–d. There was no notable damage in the *S. aureus* bacteria when in contact with the SPEEK surface. The high surface roughness and hydrophobicity of the pure PEEK and SPEEK surfaces led to the strong adhesion of the bacterial population [26].

The SPEEK-GO sample showed few bacterial attachments on the surface, as displayed in Figure 7e–g. Among those, some bacteria exhibited pores and bleb-like membrane damage, resulting in the loss of the intrinsic *S. aureus* shape and size (highlighted by the yellow arrow mark in Figure 7g). The formation of pores was thought to be due to the electrostatic tension across the membrane, which ultimately caused membrane permeabilization [45] and bacteria fatality. Interestingly, the GO sample demonstrated some biofilm resistance, probably due to the hydrophilicity and smooth surface of the SPEEK-GO film.

The surface of the bacteria-treated SPEEK-nisin sample showed limited bacterial attachment and biofilm formation in the porous structure (Figure 7h–j). Aveyard et al. found that the covalent conjugation of nisin with a polystyrene substrate significantly reduced the bacterial growth due to direct interactions with the *S. aureus* bacteria, leading to cell wall damage [5]. From the SEM micrographs of the nisin-coated SPEEK, the *S. aureus* attachment was inhibited due to the hydrophilicity of the nisin molecules, and biofilm formation was suppressed. The hydrated peptide prevented the bacteria from contact with the SPEEK-nisin sample. Even a small amount of the adhered *S. aureus* would be destroyed by the electrostatic force between the negative bacteria cell membrane and nisin’s positive surface charge.

The SPEEK-GO-nisin sample showed no *S. aureus* adhesion, as shown in Figure 7k–m. Both GO and nisin are hydrophilic and repel bacteria. Therefore, biofilm could not develop. Furthermore, a portion of nisin can be released from the conjugated surface and could damage the bacteria surrounding the peptide. The dead *S. aureus* would not be able to deposit on the SPEEK-GO-nisin surface because the full GO coverage on the SPEEK substrate would restrict bacterial adhesion, as described in the previous paragraph.

## 3. Materials and Methods

The medical grade PEEK plate (Goodfellow, Huntingdon, UK) was sliced into a square shape (5 mm × 5 mm, and with a thickness of 0.5 mm), and then utilized in this study for AMP conjugation. In order to achieve a clean PEEK surface, the square samples were bathed in acetone, deionized (DI) water, ethanol, and again DI water for 15 min each under ultrasonication at room temperature. The sample was then dried in a hot-air oven to obtain dry substrates.

### 3.1. Preparation of Sulfonated PEEK

A sulfonation method was utilized to construct the three-dimensional porous network structure on the PEEK surface. In detail, a square PEEK sample was placed in a sulfuric acid solution (98%, H_2_SO_4_, Sigma-Aldrich, St. Louis, MO, USA) for 90 s in a fumed-hood atmosphere. Afterward, the sample was removed from the acid solution and directly placed into ice-cold DI water. The remaining sulfur byproducts in the porous SPEEK were eliminated by repeatedly bathing the sample with acetone, DI water, and ethanol under ultrasonication at an ambient temperature. The SPEEK substrate was then placed in a vacuum oven at 100 °C to dry completely.

### 3.2. Preparation of SPEEK-GO Sample

The solution dip-coating method was used to prepare the SPEEK-GO sample. Initially, 2 mmol of GO (Ablonxy, Norway) was dispersed in 50 mL of DI water and ultrasonically agitated for 2 h to obtain a homogeneous dispersion solution. The dried SPEEK sample was then soaked in the GO solution for 30 min at room temperature. Afterward, the sample was removed from the solution and washed with DI water to remove the weekly grafted GO. Then, the SPEEK-GO was placed into a hot-air oven for 1 h at 100 °C. This procedure of 30 min soaking in a GO solution, washing with DI water, and then drying at 100 °C was repeated continuously three more times. Finally, the GO-grafted SPEEK sample was placed in a vacuum oven overnight at 100 °C and stored for further experiments.

### 3.3. Antimicrobial Peptide of Nisin Conjugated SPEEK-GO

The necessary quantity (0.5 wt.%) of 1-ethyl-3-(3-dimethylaminopropyl)carbodiimide (EDC, Sigma Aldrich, St. Louis, MO, USA) was first dissolved in DI water and stirred well. In this work, the nisin A peptide [ITSISLCTPGCKTGALMGCNMKTATCHCSIHVSK] was obtained from Sigma Aldrich and used as received. The nisin peptide (1 wt.%) was separately dissolved in DI water and stirred to obtain a homogeneous solution. Both the EDC and nisin solutions were slowly mixed together and continuously stirred for 40 min at ambient temperature. The SPEEK-GO plate was soaked in the nisin/EDC solution and kept in a dark atmosphere for 48 h at room temperature. Through amide linkage, the nisin peptide was conjugated to the GO using the coupling chemistry between the –NH_2_ group of the nisin and the –CO_2_H group of GO via the cross-linking reagent of EDC [32]. The SPEEK-GO-nisin was softly taken out of the nisin solution and washed with DI water to remove the residual reagents. The SPEEK-GO-nisin was then dried overnight at 40 °C in a vacuum oven. The sample was stored in a dark atmosphere at room temperature for further experiments. The nisin conjugation on a three-dimensional porous SPEEK-GO sample from a pure PEEK substrate is shown in the schematic illustration (Figure 8).

### 3.4. *Characterizations of Composite Samples*

A field emission scanning electron microscope (FESEM, SU8220, Hitachi, Tokyo, Japan) was utilized to analyze the surface micrographs of the before and after bacteria-treated SPEEK, SPEEK-GO, and SPEEK-GO-nisin samples. The elemental mapping and its chemical compositions were estimated by energy dispersive X-ray spectroscopy (EDX, XF3152, Bruker Taiwan Co. Ltd., Zhubei, Taiwan), which is attached to the FESEM instrument. The phase purity of the samples was determined by X-ray diffraction (XRD, D5005D, Siemens AG, Munich, Germany). The nisin functional groups and their conjugation effects in the SPEEK-GO were determined using attenuated total reflection-Fourier transform infrared spectroscopy (ATR-FITR, tensor 27, Bruker, Billerica, MA, USA). The sessile drop technique was used to measure the contact angle (G10MK2, Kruss GmbH, Hamburg, Germany) in order to estimate the surface wettability of the SPEEK, SPEEK-GO, and SPEEK-GO-nisin samples.

### 3.5. *Antibacterial Analysis Using an Agar Diffusion Assay*

The antibacterial behavior of the PEEK, SPEEK, SPEEK-GO, SPEEK-nisin, and SPEEK-GO-nisin samples was studied via an agar diffusion assay. This assay was used to evaluate the inhibition zones of the samples, following the Clinical & Laboratory Standards Institute (CLSI) guidelines. The agar plate was prepared with a concentration of 25 g per liter and a volume of 25 mL, which was poured into a 90 mm Petri dish. Then, a 100 µL bacterial suspension of *S. aureus* (10^6^ CFU bacterial concentration, obtained from the Bioresource Collection and Research Center, Hsinchu, Taiwan) was added onto the Petri dish along with a Luria–Bertani (LB) broth. The different SPEEK samples were gently placed on the Petri dishes containing *S. aureus* bacteria. The plates were carefully placed in an incubator at 37 °C for 24 h. The antibacterial activities of *S. aureus* was determined by measuring the diameter of inhibition zone, in millimeters (mm).

## 4. Conclusions

In summary, the three-dimensional porous SPEEK structure was obtained through the sulfonation process of the PEEK substrate. This sulfonation process resulted in a porous structure and less water wetting ability. When GO was grafted onto SPEEK, the porous surface was completely decorated with GO, leading to increased hydrophilicity. The covalent conjugating of the nisin peptide onto the SPEEK-GO surface was achieved through the EDC cross-linking pathway, thereby resulting in the lowest contact angle (24°). This nisin formed discrete islets on the flat GO surface, allowing both the GO and nisin to perform a bactericidal effect. The PEEK and SPEEK samples showed no antibacterial activity toward *S. aureus*, and this lead to significant bacterial colonization. The SPEEK-GO and SPEEK-nisin samples exhibited moderate bactericidal activities, as well as reduced the bacterial adhesion and biofilm formation when compared to the pure PEEK and SPEEK samples. The SPEEK-GO-nisin sample demonstrated the highest antibacterial activity, negligible bacterial colonization, and promising biofilm resistance. The proposed work can be applied as an active surface modification for dental and implantable materials with good antibacterial effects, biofilm resistance, and possible biocompatibility for clinical applications.

## Figures and Tables

**Figure 1 antibiotics-12-01407-f001:**
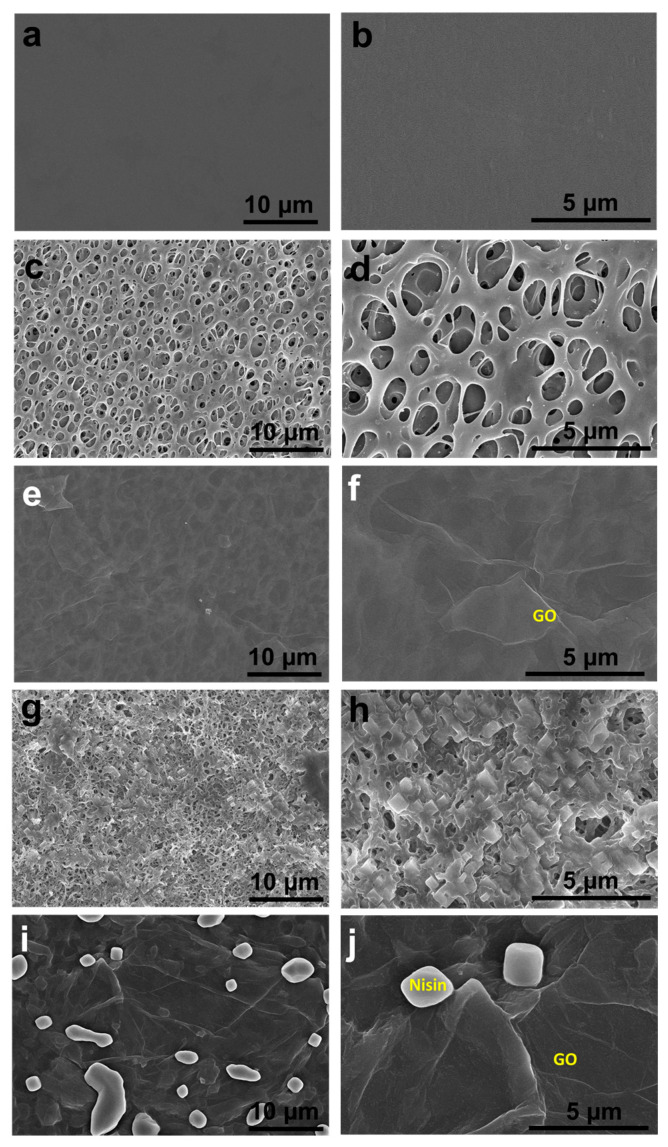
Low- and high-magnified micrographs on the surface of (**a**,**b**) PEEK, (**c**,**d**) SPEEK, (**e**,**f**) SPEEK-GO, (**g**,**h**) SPEEK-nisin, and (**i**,**j**) SPEEK-GO-nisin samples.

**Figure 2 antibiotics-12-01407-f002:**
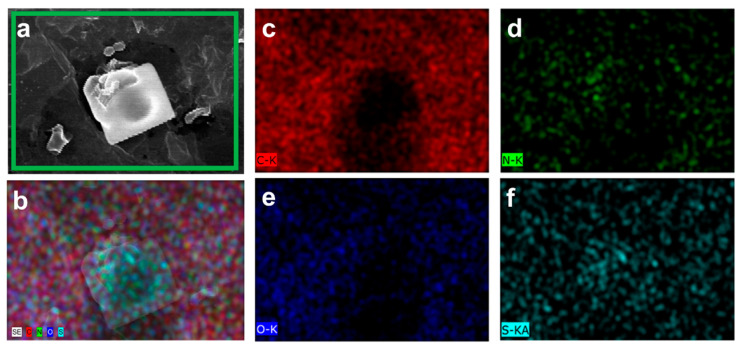
(**a**) SEM microscopic image of the surface of SPEEK-GO-nisin. Energy dispersive X-ray analysis for the mapping of (**b**) full elementals, as well as the single-elemental samples of (**c**) carbon, (**d**) nitrogen, (**e**) oxygen, and (**f**) sulfur.

**Figure 3 antibiotics-12-01407-f003:**
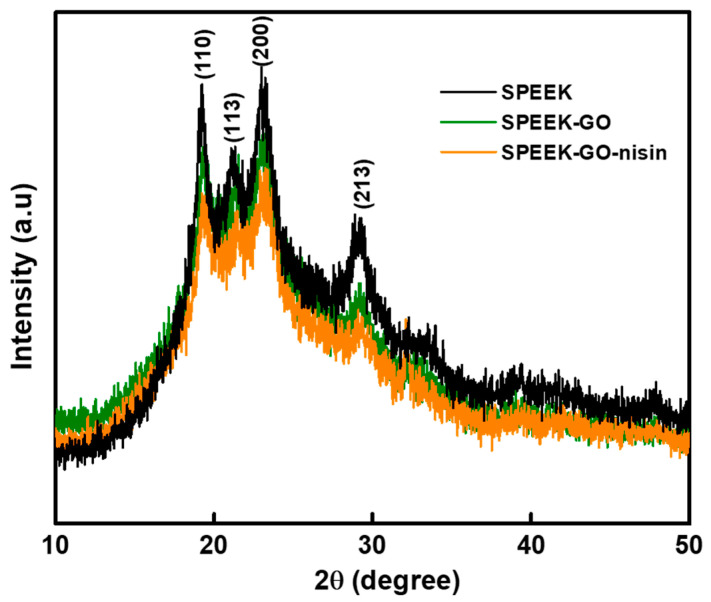
X-ray diffraction patterns of the SPEEK, SPEEK-GO and SPEEK-GO-nisin samples.

**Figure 4 antibiotics-12-01407-f004:**
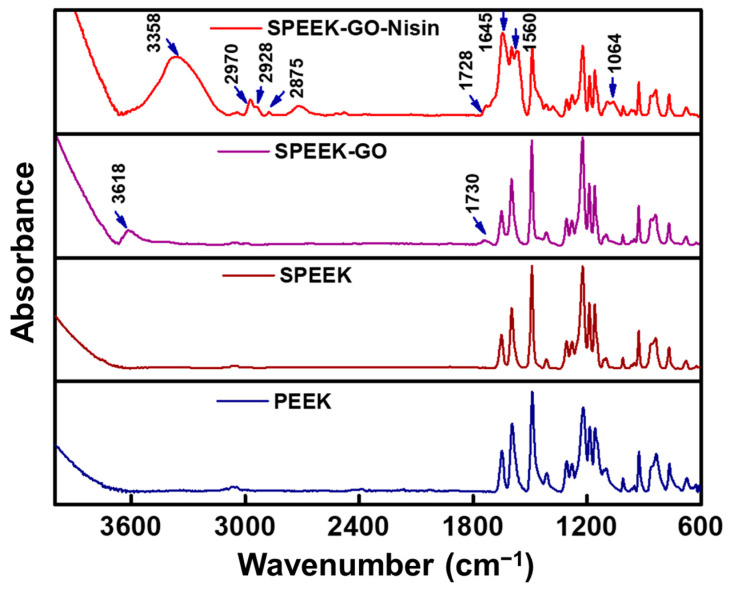
ATR-FTIR spectra of the PEEK, SPEEK, SPEEK-GO, and SPEEK-GO-nisin samples.

**Figure 5 antibiotics-12-01407-f005:**
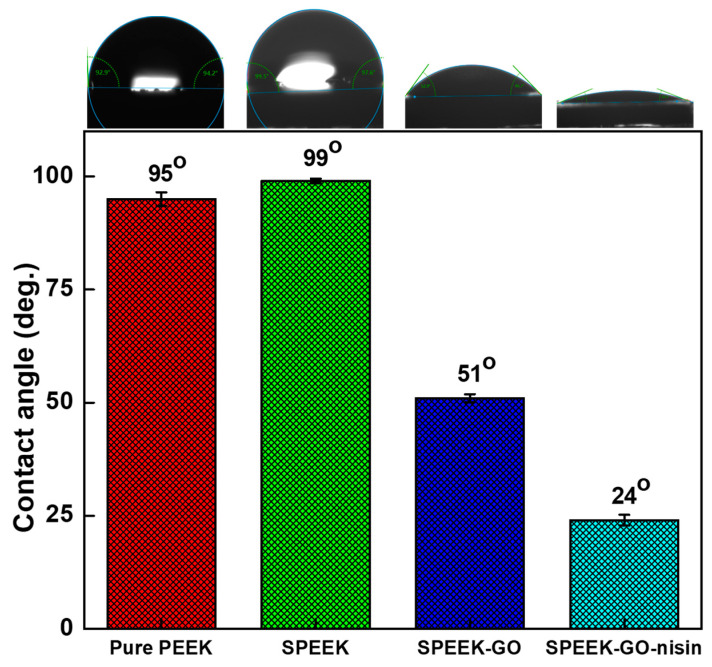
Surface contact angle analysis for the SPEEK, SPEEK-GO and SPEEK-GO-nisin substrates. Each sample surface was analyzed with at least five separate measurements. The average values are shown with standard deviations (n = 5).

**Figure 6 antibiotics-12-01407-f006:**
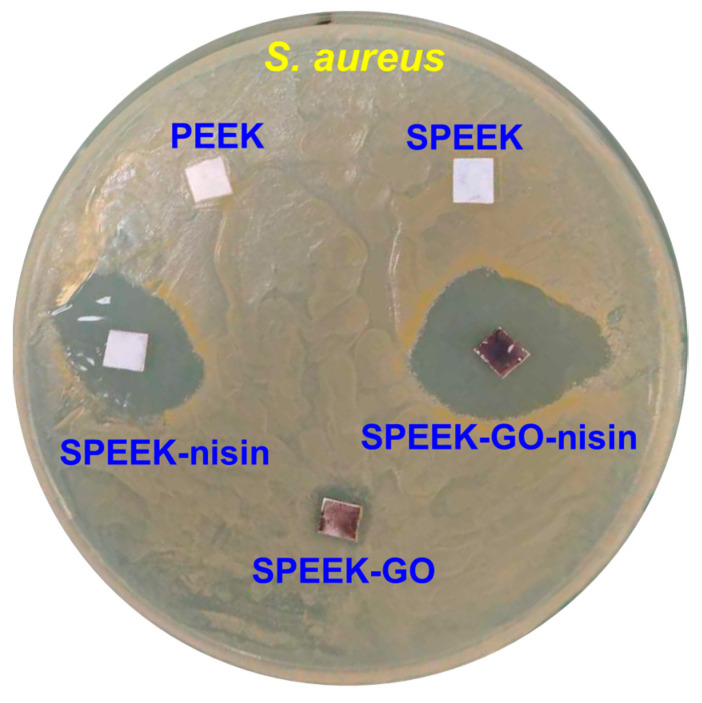
Photographic visualization of the inhibition zones against *S. aureus* bacteria when using the SPEEK, SPEEK-GO, SPEEK-nisin, and SPEEK-GO-nisin samples. The inhibition zone diameter values are calculated with three replicates (n = 3).

**Figure 7 antibiotics-12-01407-f007:**
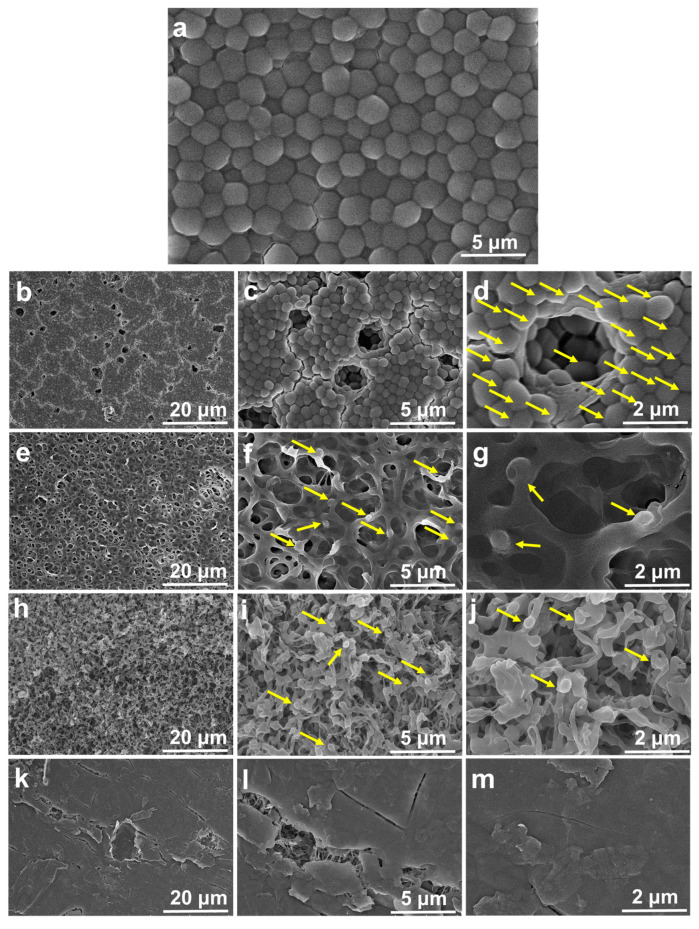
Surface micrographs of low and high magnifications for *S. aureus* when exposed to the (**a**) pure PEEK, (**b**–**d**) SPEEK, (**e**–**g**) SPEEK-GO, (**h**–**j**) SPEEK-nisin, and (**k**–**m**) SPEEK-GO-nisin samples. Arrows indicate the presence of *S. aureus* bacteria.

**Figure 8 antibiotics-12-01407-f008:**
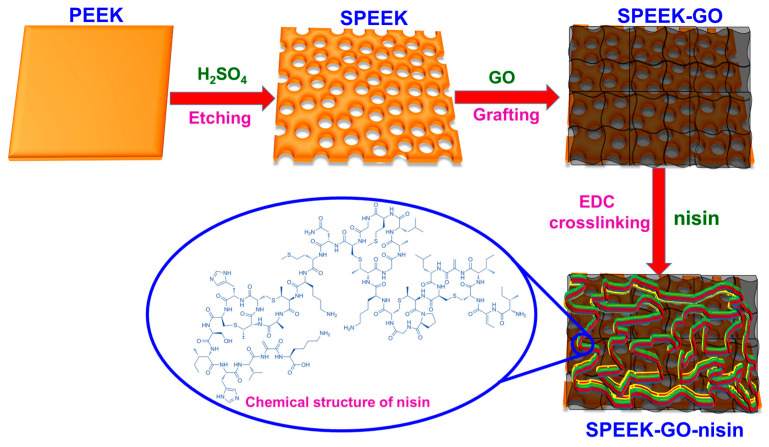
Schematic illustration of the acid etching preparation for the GO grafting and SPEEK-GO-nisin samples from the pure PEEK substrate.

## Data Availability

The data presented in this work are available on request from the corresponding authors.

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
