# Peer review of "Antimicrobial Peptide Conjugated on Graphene Oxide-Containing Sulfonated Polyetheretherketone Substrate for Effective Antibacterial Activities against Staphylococcus aureus"

_antibiotics, 2023, doi:10.3390/antibiotics12091407_

Round 1
Reviewer 1 Report
This investigation, submitted by S. J. Lue and co-workers, describe the synthesis and analysis of SPEEK-GO-nisin materials. The successful nisin conjugation onto the SPEEK-GO surface was confirmed by surface micrographs, X-ray diffraction analysis, and ATR-FTIR spectroscopy. Antibacterial activities were further studied against S. auereus bacteria, thereby showing a clear antibiofilm effect of the SPEEK-GO-nisin composite.
After careful reading, I recommend reconsidering this work after major revisions have been performed.
Actually, the paper is quite well written, although a moderate editing of the English language is required. The work is significant, properly organized, and it is interesting for the scientific community. However, I have a major concern regarding the determination and quantification of the nisin that has been grafted onto the SPEEK-GO surface. There are not any adsorption isotherms models (Langmuir, Freundlich, etc.).
Except this major point, I further recommend minor revisions for a better reading of this paper.
Ref. 1-3 are not very appropriate.
Fig. 6. See caption: zone against E. Coli (left side)???
L253-L255: I am not sure it is a good interpretation. Regarding ref. 41, it just means physically adsorbed SPEEK-GO-nisin may have even better antibacterial activities than covalently bonded one.
References: Please use the good template for references = Journal names must be abbreviated, and issue n° is not shown. Furthermore, the volume must be italicized.
Moderate editing of English language is required
Author Response
Review 1
(x) I would not like to sign my review report
( ) I would like to sign my review report
( ) I am not qualified to assess the quality of English in this paper
( ) English very difficult to understand/incomprehensible
( ) Extensive editing of English language required
(x) Moderate editing of English language required
( ) Minor editing of English language required
( ) English language fine. No issues detected
Response: The English language is edited by professional language editor.
Comments and Suggestions for Authors
This investigation, submitted by S. J. Lue and co-workers, describe the synthesis and analysis of SPEEK-GO-nisin materials. The successful nisin conjugation onto the SPEEK-GO surface was confirmed by surface micrographs, X-ray diffraction analysis, and ATR-FTIR spectroscopy. Antibacterial activities were further studied against S. auereus bacteria, thereby showing a clear antibiofilm effect of the SPEEK-GO-nisin composite. After careful reading, I recommend reconsidering this work after major revisions have been performed.
Response: Thank you very much for your detailed and positive comments.
Q1. Actually, the paper is quite well written, although a moderate editing of the English language is required. The work is significant, properly organized, and it is interesting for the scientific community. However, I have a major concern regarding the determination and quantification of the nisin that has been grafted onto the SPEEK-GO surface. There are not any adsorption isotherms models (Langmuir, Freundlich, etc.). Except this major point, I further recommend minor revisions for a better reading of this paper.
Response: As shown in Fig. 1, the nisin was grafted in isolated islet form on flat GO surface. Therefore, the adsorption isotherm is not applicable. The new data on EDX analysis was studied for all samples and the nisin grafting level was included in the 3rd paragraph of 2.1, p. 5.
Q2. Ref. 1-3 are not very appropriate.
Response: Thank you for pointing out this. We revised the corresponding references.
Q3. Fig. 6. See caption: zone against E. Coli (left side)???
Response: Thank you for pointing this out. The typo error is corrected and the caption is revised, p.9.
Q4. L253-L255: I am not sure it is a good interpretation. Regarding ref. 41, it just means physically adsorbed SPEEK-GO-nisin may have even better antibacterial activities than covalently bonded one.
Response: The sentence and reference were deleted to avoid confusion.
Q5. References: Please use the good template for references = Journal names must be abbreviated, and issue n° is not shown. Furthermore, the volume must be italicized.
Response: Thank you for pointing this out. All the references are revised based on the MDPI journal formats.

Reviewer 2 Report
Firstly, I would like to acknowledge the authors' efforts in addressing an important and timely research question. The manuscript presents a comprehensive analysis of an important peptide family nisin. The study design, data collection, and statistical analysis appear to be appropriate and well-described.
However, the manuscript is not acceptable in its current form. There are some major concerns about the methodology and the controls that were selected for the experiments. They need to be addressed before it get accepted in the Antibiotics journal.
- Please introduce your peptide properly. Nisin Peptide is in various forms. Like Nisin Z, Nisin A, Nisin Q, Nisin J, Nisin P, Nisin S, Nisin H etc. The authors never mentioned which Nisin peptide they worked on. Let alone providing the sequence. Please describe the peptide first before discussing conjugation. Also please provide the sequence of the peptide in the very first figure.
- Nisin peptides are themselves antimicrobial. In Figure 6, why Nisin peptide alone was not taken as a control to truly evaluate the antibacterial potential of the conjugate? Please provide the zone of inhibition of the peptide alone to truly compare the level of antibacterial activity.
- Similarly in Figure 7, please provide the data of peptide alone
- Please provide the toxicity data for all the conjugations and the peptide. This is important to truly evaluate the antibacterial activity of the peptide. Hemolytic data or MTT data with normal fibroblast such as 3T3 should be sufficient.
In conclusion, while the manuscript addresses an important research question and exhibits several strengths, there are significant areas that require major revision. I strongly believe that with careful attention to the concerns outlined above, the manuscript has the potential to become a contribution to the field.
I recommend that the authors be given the opportunity to revise and resubmit the manuscript. Given the nature of the required revisions, it would be appropriate that major revisions be made.
Author Response
Reviewer 2
( ) I would not like to sign my review report
(x) I would like to sign my review report
( ) I am not qualified to assess the quality of English in this paper
( ) English very difficult to understand/incomprehensible
( ) Extensive editing of English language required
( ) Moderate editing of English language required
( ) Minor editing of English language required
(x) English language fine. No issues detected
Response: Thank you very much for your positive comments.
Comments and Suggestions for Authors
Firstly, I would like to acknowledge the authors' efforts in addressing an important and timely research question. The manuscript presents a comprehensive analysis of an important peptide family nisin. The study design, data collection, and statistical analysis appear to be appropriate and well-described. However, the manuscript is not acceptable in its current form. There are some major concerns about the methodology and the controls that were selected for the experiments. They need to be addressed before it get accepted in the Antibiotics journal.
Response: Thank you very much for your detailed and positive comments.
Q1. Please introduce your peptide properly. Nisin Peptide is in various forms. Like Nisin Z, Nisin A, Nisin Q, Nisin J, Nisin P, Nisin S, Nisin H etc. The authors never mentioned which Nisin peptide they worked on. Let alone providing the sequence. Please describe the peptide first before discussing conjugation. Also please provide the sequence of the peptide in the very first figure.
Response: In this work, nisin A peptide was obtained from Sigma Aldrich and directly used without any sequence alternation. The sequence of the nisin is included in the 1st paragraph of the experimental section 3.3, p.12. The chemical structure is also included in schematic Fig. 8.
Q2. Nisin peptides are themselves antimicrobial. In Figure 6, why Nisin peptide alone was not taken as a control to truly evaluate the antibacterial potential of the conjugate? Please provide the zone of inhibition of the peptide alone to truly compare the level of antibacterial activity.
Response: We agree with reviewer. The new experiment on control data of nisin-coated SPEEK was studied, and the results and discussion were included in the sections 2.3 and 2.4.
Q3. Similarly in Figure 7, please provide the data of peptide alone.
Response: The morphological analysis of the SPEEK-nisin is shown in Fig. 1g,h. The SPEEK-nisin loaded with bacterial is analyzed (Fig. 7) and included in the revised manuscript. The corresponding discussion was included in the sections 2.1& 2.4.
Q4. Please provide the toxicity data for all the conjugations and the peptide. This is important to truly evaluate the antibacterial activity of the peptide. Hemolytic data or MTT data with normal fibroblast such as 3T3 should be sufficient.
Response: Thank you for your suggestion. The nisin A is an FDA approved food material with no major toxicity effect. The new sentence and its corresponding references are included in the last paragraph of section 2.3, p.9.
In conclusion, while the manuscript addresses an important research question and exhibits several strengths, there are significant areas that require major revision. I strongly believe that with careful attention to the concerns outlined above, the manuscript has the potential to become a contribution to the field. I recommend that the authors be given the opportunity to revise and resubmit the manuscript. Given the nature of the required revisions, it would be appropriate that major revisions be made.
Response: Thank you very much for your detailed and positive comment.

Reviewer 3 Report
The manuscript entitled „Antimicrobial peptide conjugated graphene oxide containing 2 sulfonated polyetheretherketone substrate for effective antibac-3 terial activities against Staphylococcus aureus”, submitted for evaluation to Antibiotics, presents the evaluation of the properties of PEEK polymer modified by acid etching, GO binding and nisin coupling, to increase the nonspecific antimicrobial activity, in future treatment of bone infections.
This work is clear and nicely constructed, written in good and comprehensible English. The hypothesis is clear and supported by results.
I have only some comments and I found some errors. My comments and questions concerning the submitted article are listed below:
COMMENTS TO AUTHORS
1. Please insert the wavelengths of the peaks marked in Figure 4. Moreover, the absorption area between 800 and 3400 cm -1 cannot be named “absorption band” because it consists of numerous bands. Please correct it.
2. “kanchanapally” is a surname and should be written in capital letter.
3. Please provide the amount of EDC used in the experiment (section 3.3.) and density of bacterial inoculate (section 3.5.), a s well as agar concentration in agar-solidified medium.
Author Response
Reviewer 3
( ) I am not qualified to assess the quality of English in this paper
( ) English very difficult to understand/incomprehensible
( ) Extensive editing of English language required
( ) Moderate editing of English language required
( ) Minor editing of English language required
(x) English language fine. No issues detected
Comments and Suggestions for Authors
The manuscript entitled, Antimicrobial peptide conjugated graphene oxide containing sulfonated polyetheretherketone substrate for effective antibacterial activities against Staphylococcus aureus”, submitted for evaluation to Antibiotics, presents the evaluation of the properties of PEEK polymer modified by acid etching, GO binding and nisin coupling, to increase the nonspecific antimicrobial activity, in future treatment of bone infections. This work is clear and nicely constructed, written in good and comprehensible English. The hypothesis is clear and supported by results. I have only some comments and I found some errors. My comments and questions concerning the submitted article are listed below:
Response: Thank you very much for your detailed and positive comments.
COMMENTS TO AUTHORS
Q1. Please insert the wavelengths of the peaks marked in Figure 4. Moreover, the absorption area between 800 and 3400 cm-1 cannot be named “absorption band” because it consists of numerous bands. Please correct it.
Response: Thank you for pointing this out. The typo error is corrected in the 3rd paragraph of section 2.2 and the peak wavelength is marked in Fig. 4.
Q2. “kanchanapally” is a surname and should be written in capital letter.
Response: Thank you for pointing this out. The typo error is corrected.
Q3. Please provide the amount of EDC used in the experiment (section 3.3.) and density of bacterial inoculate (section 3.5.), as well as agar concentration in agar-solidified medium.
Response: Thank you for pointing this out. The details are included in the experimental sections 3.3 and 3.5.

Round 2
Reviewer 2 Report
The manuscript requires substantial revisions, with particular emphasis on addressing the significant concerns that have been either overlooked or intentionally disregarded.
Regarding Figures 6 and 7, I specifically requested information regarding the zone of inhibition attributable to the peptide ALONE. However, the authors chose to focus on the topic of conjugation, sidestepping the requested details.
Similarly, when I inquired about the toxicity data for the conjugate, the authors responded by discussing the toxicity of the peptide in isolation.
To facilitate the acceptance of the manuscript in the journal, I kindly request that the aforementioned data be provided as initially requested. This will ensure that the manuscript aligns more closely with the journal's requirements and addresses the concerns that have been raised.
Author Response
The manuscript requires substantial revisions, with particular emphasis on addressing the significant concerns that have been either overlooked or intentionally disregarded.
Q1. Regarding Figures 6 and 7, I specifically requested information regarding the zone of inhibition attributable to the peptide ALONE. However, the authors chose to focus on the topic of conjugation, sidestepping the requested details.
Response: We think this request is rebuttable. The antibacterial efficacy of the pure nisin against S. aureus is documented in the literature [Inter. J. Food Microbio. 2015, 211, 38-43; Front. in Microbio. 2020, 11, 1007; Food Control 2016, 59, 499-506] and cited in this article (p. 3). There is no need to repeat the nisin peptide alone test. That result, even carried out, cannot be compared with the coated samples due to the different diffusion and release rates from the liquid suspension (peptide alone) and SPEEK coated surface. Therefore, we included the new data on SPEEK/nisin antibacterial test and post-bacterial FESEM analysis for fair comparison (as revised in Fig. 6&7).
Q2. Similarly, when I inquired about the toxicity data for the conjugate, the authors responded by discussing the toxicity of the peptide in isolation. To facilitate the acceptance of the manuscript in the journal, I kindly request that the aforementioned data be provided as initially requested. This will ensure that the manuscript aligns more closely with the journal's requirements and addresses the concerns that have been raised.
Response: We think this request is rebuttable. The biocompatibility of the nisin is reported and well cited in the manuscript (last paragraph of section 2.3, p. 9). That issue is out of the scope of this research and this special issue. Therefore, we don’t think it is suitable to include that result.
Round 3
Reviewer 2 Report
The paper can be accepted.